# Double-µPeriscope, a tool for multilayer optical recordings, optogenetic stimulations or both

**Mototaka Suzuki[1]\*[†], Jaan Aru[1,2], Matthew E Larkum[1]**

[1]Institute of Biology, Humboldt University of Berlin, Berlin, Germany; [2]Institute of Computer Science, University of Tartu, Tartu, Estonia

**Abstract** Intelligent behavior and cognitive functions in mammals depend on cortical microcircuits made up of a variety of excitatory and inhibitory cells that form a forest-like complex across six layers. Mechanistic understanding of cortical microcircuits requires both manipulation and monitoring of multiple layers and interactions between them. However, existing techniques are limited as to simultaneous monitoring and stimulation at different depths without damaging a large volume of cortical tissue. Here, we present a relatively simple and versatile method for delivering light to any two cortical layers simultaneously. The method uses a tiny optical probe consisting of two microprisms mounted on a single shaft. We demonstrate the versatility of the probe in three sets of experiments: first, two distinct cortical layers were optogenetically and independently manipulated; second, one layer was stimulated while the activity of another layer was monitored; third, the activity of thalamic axons distributed in two distinct cortical layers was simultaneously monitored in awake mice. Its simple-design, versatility, small-size, and low-cost allow the probe to be applied widely to address important biological questions.

**\*For correspondence:**
mototaka@gmail.com

**Present address:** [†]Swammerdam Institute for Life Sciences, University of Amsterdam, Amsterdam, Netherlands

**Competing interest:** The authors declare that no competing interests exist.

## Editor's evaluation

This Tools and Resources article details an innovative method for simultaneously stimulating and imaging two cortical layers in tandem while causing minimal damage to brain tissue. The method substantially builds on existing methods in several ways, while still pinpointing the limitations of existing methods that are overcome in this new approach. Three well-described sets of experiments demonstrate the method's reliability and versatility, and highlight its promise in tackling big questions about cortical microcircuit functions.

## Introduction

Despite over half a century of intense investigation, we still do not understand the role of layers in the complicated microcircuitry of the cortex (*Adesnik and Naka, 2018*). Over the last decade optogenetic manipulation – either activation or inhibition – of specific cells (*Nagel et al., 2003*; *Boyden et al., 2005*) has been established as a powerful method to examine the causal relationship between the specific cell type and the network activity, cognitive functions or animal behavior (*Deisseroth, 2015*). Multichannel electrical recording is the most widely used approach to measure the neural responses to optogenetic stimulation. This approach is preferred over the optical approaches available (e.g., two-photon imaging) because of the relative ease of use and lower costs. The problem with this approach, however, is that extracellular signals represent the summation of the activity of all structures (dendrites, axons, and cell bodies) from all cell types surrounding each electrode; even worse, extracellular activity is usually very complex in vivo, particularly in the awake brain (*Buzsáki,*

*2006*). It is extremely difficult to explain why and how every single pattern of extracellular activities is generated. Polysynaptic activations can also occur through local and long-range connections. It is therefore necessary to consider feedback projections with various delays, which makes it further difficult to disentangle distinct contributions of different neuronal structures to extracellular signals.

One approach to overcome this problem is to use two-photon microscopy and all-optical approach with two light sources – one for optogenetic stimulation and the other for imaging – and visualize how stimulation of specific cell-type evokes responses in surrounding neurons (*Packer et al., 2015*). Though most approaches are either limited in range (i.e., the total depth) or speed, recently developed methods permit optogenetic stimulation or calcium imaging of ~50 neurons that are arbitrarily distributed in three-dimensional space (*Pégard et al., 2017*; *Geiller et al., 2020*) as well as fast volumetric cortex-wide two-photon imaging at cellular resolution (*Demas et al., 2021*; *Prevedel et al., 2016*). Imaging across different cortical depths (*z*-axis) is more difficult to implement than the more conventional *x–y* scanning.

More invasive approaches using a large (1 mm) prism covering all cortical layers would allow simultaneous imaging to neurons across layers (*Chia and Levene, 2009*; *Andermann et al., 2013*); however, the significant damage caused by 1 mm prism insertion into the cortical tissue must be considered. Substantial bleeding during surgery is unavoidable and a significant number of cells are killed or damaged by the procedure (*Andermann et al., 2013*), suggesting the possibility that repairing and rewiring processes take place, even if the animal survives the injury. Since the repairing process may significantly alter the local and long-range connections between survived neurons, the imaged neuronal network may no longer function in a way the normal healthy network did before the operation. Moreover, this approach typically requires chronic surgery involving a substantial delay between the surgery and recordings. Less invasive approaches use a single tapered fiber that has multiple windows (*Pisanello et al., 2014*) or micro-LED probes (*Cao et al., 2013*; *McAlinden et al.,*

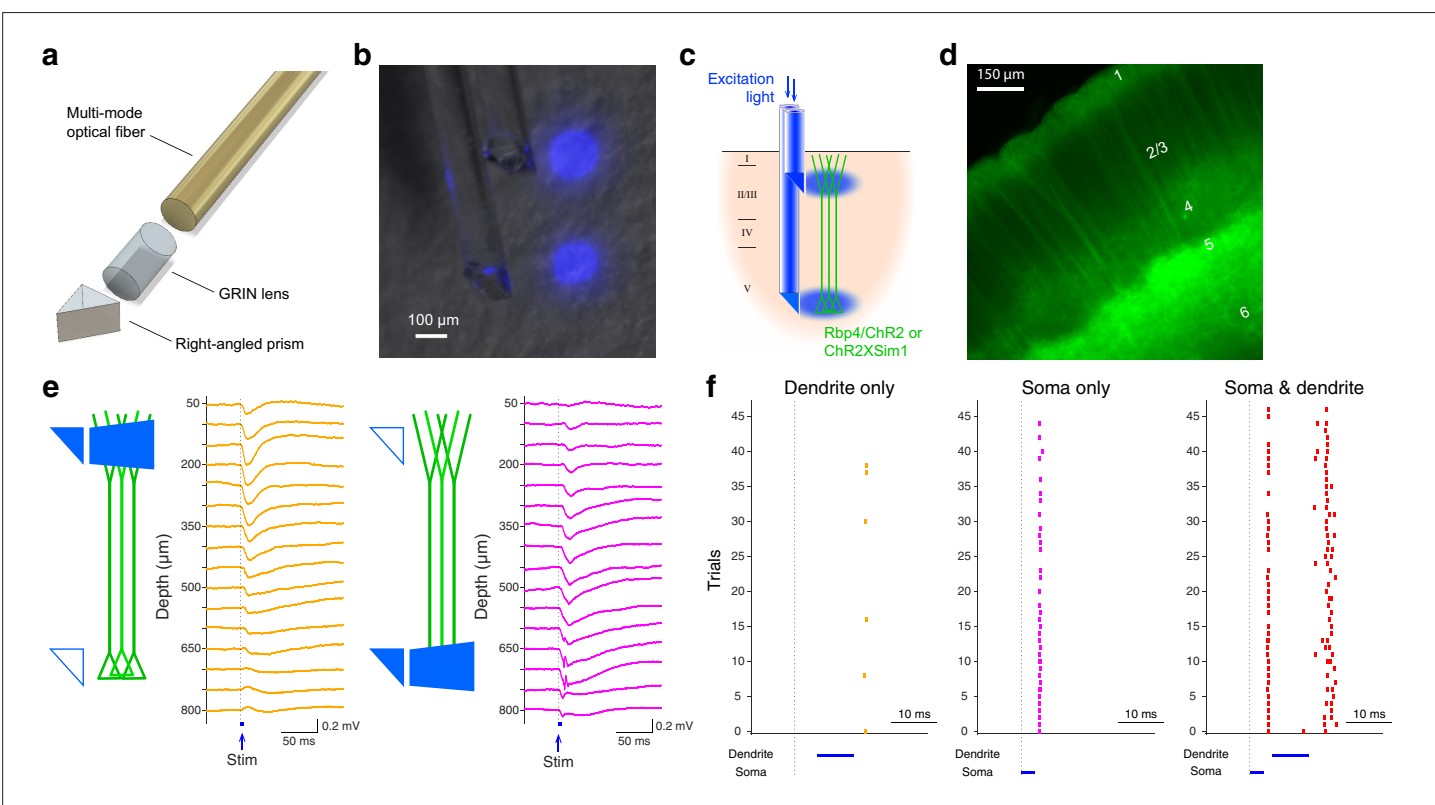

**Figure 1.** The design and functionality of double-µPeriscope. (**a**) An exploded view of an individual µPeriscope. (**b**) A photomicrograph of the double-µPeriscope with two blue light spots. (**c**) Schematic diagram of the experiment. Rbp4-Cre mice virally expressing channelrhodopsin 2 (ChR2) or mice in Sim1-Cre line crossed with ChR2 reporter line (Ai32) were used. (**d**) A photomicrograph of the cortical slice where layer 5 (L5) pyramidal neurons express ChR2 and yellow fluorescent protein (YFP). (**e**) Evoked potentials by dendritic stimulation (left) versus somatic stimulation (right). (**f**) Somatic action potentials (APs) evoked by dendritic stimulation only (left), somatic stimulation only (middle) and both (right). Vertical ticks denote APs.

2015; *Kim et al., 2020*); however, they are only capable of optogenetic stimulation and do not permit calcium fluorescence imaging.

A relatively standard approach is the insertion of an endoscope that with the aid of a right-angled microprism can deliver light to and receive light from a single cortical layer simultaneously (*Suzuki and Larkum, 2017*; *Suzuki and Larkum, 2020*). To take advantage of the ease and flexibility of the single endoscope approach while allowing the probing of multiple layers, we developed a double-µPeriscope that is capable of optogenetic stimulation, fluorescence imaging or both of two separate cortical layers (*Figure 1a, b*). Each µPeriscope consists of a 0.10 mm × 0.10 mm or 0.18 mm × 0.18 mm micro right-angled prism, a custom-designed gradient-index (GRIN) lens, and a multimode optical fiber. To examine the focal light delivery from each µPeriscope, we stimulated two compartments of cortical layer 5 (L5) pyramidal neurons by inserting the double-µPeriscope with the lower one in cortical L5 and the upper one in layer 1 (L1) (*Figure 1c*). We expressed light-sensitive ion channels, channelrhodopsin 2 (ChR2) (*Nagel et al., 2003*; *Boyden et al., 2005*) and yellow fluorescent protein (YFP) in L5 pyramidal neurons (*Figure 1d*). A double-µPeriscope allowed us to address a wide range of neurobiological problems, three of which are described below.

## Results

First, a previous in vitro study showed that combined stimulation of cell body and distal apical dendrites within a close time window induces a large, regenerative, long-lasting calcium plateau potential that evokes a burst somatic action potentials (APs) (*Larkum et al., 1999*) – a phenomenon referred to as 'backpropagation-activated calcium spike firing' or BAC firing. However, due to technical difficulties, it remains unknown whether and under what conditions BAC firing occurs in vivo. The double-µPeriscope, in combination with electrophysiological recordings from L5 allowed us to directly examine this question in vivo for the first time. To express ChR2 in L5 pyramidal neurons, a transgenic mouse line Sim1-KJ18-Cre × Ai32 was used. In agreement with the previous in vitro data, somatic L5 stimulation preceding dendritic L1 stimulation in vivo under anesthesia evoked more prominent deflections of the local field potential (*Figure 1e*) and more APs (*Figure 1f*), compared to the stimulation of either somatic or distal dendritic compartment alone. This result suggests that the backpropagating AP-induced $Ca^{2+}$ spike firing does occur also in vivo. This finding highlights the benefits of the double-µPeriscope.

The same double-µPeriscope is capable of studying the interaction between cell classes in different layers. As an example, we examined the effect of optogenetic stimulation of cortical layer 2/3 (L2/3) on L5 (*Figure 2a*) by expressing ChR2 and YFP in L2/3 pyramidal cells in the primary motor cortex (*Figure 2b*) through combined use of a transgenic mouse line Rasgrf2-2A-dCre with Cre-dependent adeno-associated virus vector with CaMKIIα promotor. The upper µPeriscope stimulated ChR2-expressing L2/3 pyramidal cells while the lower µPeriscope measured the calcium fluorescence in L5 where a cell-permeable synthetic calcium indicator (Cal-590 AM) was preinjected. In agreement with the recent study in primary somatosensory area using a different approach (electrophysiological recordings) (*Pluta et al., 2019*), the net effect of L2/3 stimulation onto L5 turned out to be inhibitory in the motor cortex (*Figure 2c, d*). This result suggests that the L2/3-to-L5 inhibition might be a general principle across the cortex in contrast to the prevailing textbook assumption that L2/3-to-L5 projections are excitatory (*Douglas and Martin, 2004*). The previous study suggested that somatostatin-positive interneurons are likely to be responsible for the translaminar inhibition in the somatosensory cortex (*Pluta et al., 2019*); whether the same mechanism mediates the translaminar inhibition in the primary motor cortex needs more investigations that are beyond the scope of this technical report.

Lastly, we demonstrate that a double-µPeriscope can be applied for simultaneous fluorescence imaging of axonal terminals distributed at two depths (*Figure 2e*). Compared to cell bodies and dendrites, two-photon imaging of axonal activities at deep layers is still challenging (*Broussard et al., 2018*), even at a single focal plane, let alone fast sequential imaging at multiple depths. The postero-medial thalamic nucleus (POm) sends axons to cortical layers 1 and 5a in the primary somatosensory area (*Figure 2f*; *Wimmer et al., 2010*); however, it is unknown whether the activities of these two axonal pathways are same or different. This is important for ascertaining if the POm delivers the same or different information to these different layers which in turn is important for describing the functional role of higher-order thalamic input to the cortex. Using the double-µPeriscope we measured the axonal activities in cortical layers 1 and 5a simultaneously (*Figure 2e*) in the awake head-fixed

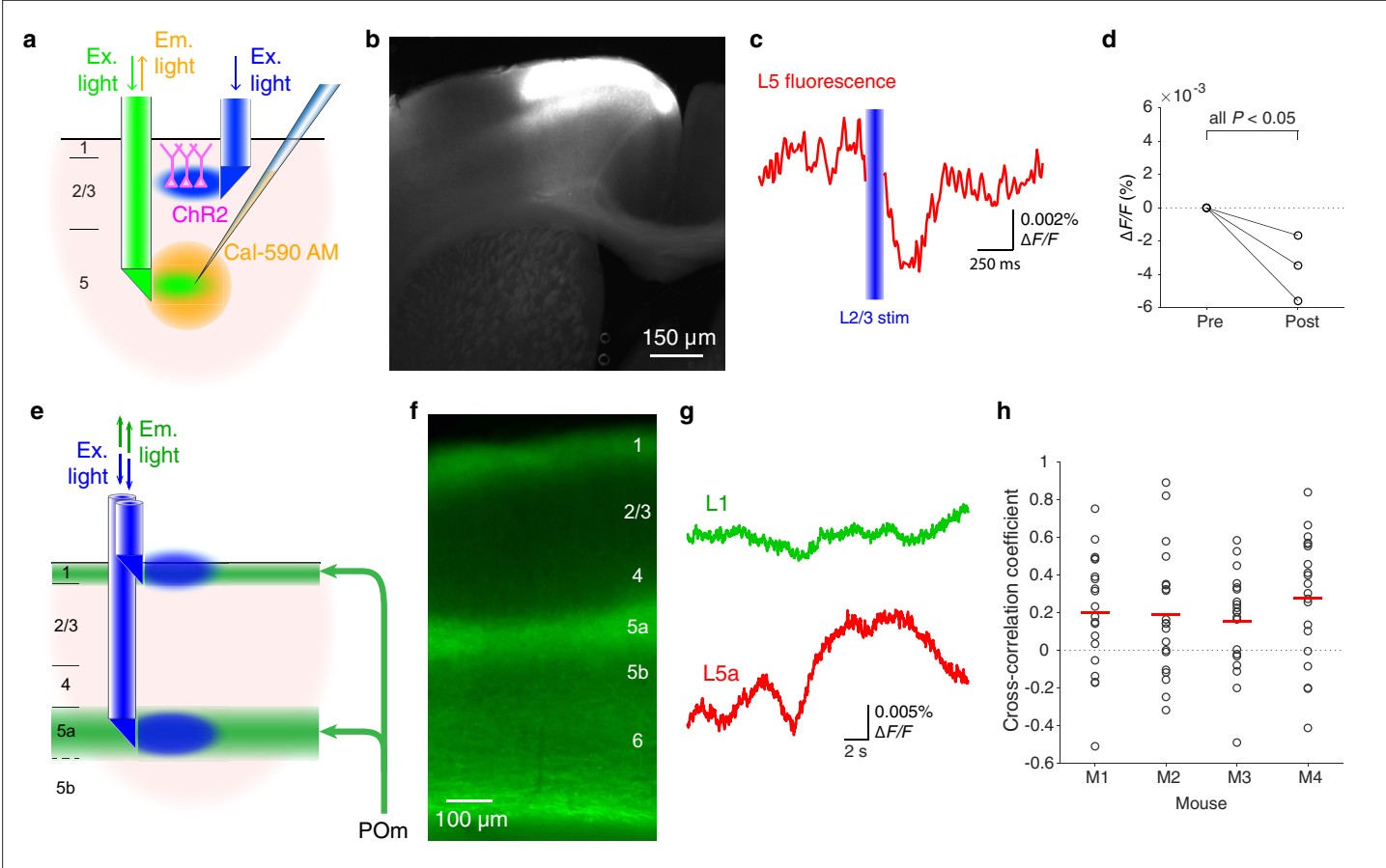

**Figure 2.** Additional functionalities of double-µPeriscope. (**a**) Schematic diagram of the experiment that combines optogenetic stimulation and calcium fluorescence imaging. (**b**) A photomicrograph of cortical slice where L2/3 pyramidal cells express channelrhodopsin 2 (ChR2) and yellow fluorescent protein (YFP) in the primary motor cortex. (**c**) Fluorescence change in L5 caused by L2/3 stimulation. The trace is the average over 20 measurements. (**d**) Summary of data from Rasgrf2-2A-dCre mice where the fluorescence reduction in L5 after optogenetic stimulation of L2/3 was statistically significant ($n$ = 3 mice, 20 measurements in each mouse, all p < 0.05, two-tailed Student's $t$-test). (**e**) Schematic diagram of the experiment where population fluorescence of thalamic axons distributed in cortical layers 1 and 5a was simultaneously measured. (**f**) A photomicrograph of cortical slice where axonal terminals from POm are densely distributed in cortical layers 1 and 5a. (**g**) Example traces of simultaneously measured fluorescence in cortical layers 1 (green) and 5a (red). (**h**) Summary of cross-correlation coefficients obtained from four Gpr26-cre mice. An open circle indicates a pair of simultaneous measurements of axonal activities in layers 1 and 5a. Red line indicates the mean of 20 measurements in each mouse. Ex, excitation; Em, emission.

mice undergoing whisker stimulation. A genetic calcium indicator GCaMP6s (*Chen et al., 2013*) was expressed in POm using a transgenic mouse line Gpr26-cre (*Gong et al., 2007*) and a Cre-dependent adeno-associated virus vector encoding GCaMP6s. We found that the axonal activities in layers 1 and 5a are surprisingly variable; they are clearly uncorrelated in some trials but highly correlated in others (*Figure 2g, h*). The cross-correlation analysis confirmed this variability, although the mean coefficient was lower than 0.3 in all four mice (*Figure 2h*), suggesting the possibility that POm neurons deliver different types of information to layers 1 and 5a. Though further studies are necessary to explain the large variability of axonal activity in layers 1 and 5a, this new tool will contribute to our understanding of these important thalamocortical pathways and their role in neural computations and cognition (*Mease et al., 2016*; *Audette et al., 2018*; *Aru et al., 2019*; *Aru et al., 2020*). These three sets of preliminary experiments demonstrate that the new tool provides us with a new approach for addressing previously intractable important neurobiological problems.

## Discussion

Multiple bare optical fibers with a flat end have been used to image multiple brain regions (*Kim et al., 2016*; *Sych et al., 2019*). However, this approach is suitable if the target brain areas are sufficiently

separate (>1 mm), but as shown in the three sets of experiments, our method has significant advantages to investigate more close (<1 mm) structures such as cortical layers in the rodent cortex. Optical fibers with a flat end would need higher pressure and therefore damage the cortical tissue upon insertion. In contrast, the sharp edge of microprism allows us to smoothly insert the probe with significantly less damage. Moreover, layer-specific activation/imaging with a downward oriented light beam from a bare optical fiber is problematic; in principle it is possible to calibrate the light power to optimize the signal from a single layer versus the layers below, but in practice this is always an approximation and it is impossible to be sure in any given preparation exactly how much signal is derived from the desired layer versus the others. Furthermore, it involves reducing the power of the excitation light considerably which leads to signal-to-noise issues.

Previous studies used an electric lens combined with two-photon microscopy (*Grewe et al., 2011*; *Beaulieu-Laroche et al., 2019*; *Francioni et al., 2019*) to measure the cellular activities at two depths. The limitations of this approach are (1) that imaging is not simultaneous because the electric shift of focal plane takes time for shifting and stabilization; and (2) that the maximum distance of shift is 500 µm; therefore, the distance between two structures under study must be within this range. A related issue is that imaging deeper structures needs more laser power. Therefore, even if the lens is capable of shifting the focal plane by a larger distance (than 500 µm), the excitation light must be rapidly switched between high and low power in perfect synchronization with the electrical shift of focal plane, which is technically and financially challenging in most laboratories. Besides, deep subcortical structures cannot be imaged with this approach unless a substantial amount of overlaying cortical tissue is surgically removed (*Dombeck et al., 2010*). The double-µPeriscope we show here does not have these limitations because the distance between two µPeriscopes can be freely adjusted, and imaging is simultaneous no matter how distant the two µPeriscopes are. Deep subcortical structures can also be imaged or optogenetically stimulated with the double-µPeriscope without substantial damage of cortical tissue.

Importantly, the cost to setup the imaging system with a double-µPeriscope is an order of magnitude less expensive. The reason for the significantly lower cost is that it requires only tiny prisms, GRIN lenses and a high sensitivity scientific camera as the main components. Given ever-improving scientific cameras and lowering costs of micro-optics that can now be 3D printed (*Gissibl et al., 2016*), approaches using micro-optics such as presented here are likely to be more appealing for many laboratories. We demonstrated its versatility through three sets of experiments where a double-µPeriscope can be used for either optogenetic stimulation, calcium fluorescence imaging, or both of two closely separate cortical layers. The present version of double-µPeriscope uses single-core multimode fiber; therefore, it is impossible to visualize subcellular structures in the imaged layer. However, a commercially available multicore optical fiber bundle would allow this if spatial resolution is required in other applications.

# Materials and methods

## Key resources table

| Reagent type (species) or resource | Designation | Source or reference | Identifiers | Additional information |
|---|---|---|---|---|
| Genetic reagent (*Mus musculus*) | Short: Sim1-KJ18-Cre *Tg(Sim1-cre)KJ18Gsat* | *Gerfen et al., 2013* | RRID:MMRRC_031742-UCD | |
| Genetic reagent (*Mus musculus*) | Short: Ai32 *Ai32(RCL-ChR2(H134R)/EYFP)* | *Madisen et al., 2012* | RRID:Addgene_34880 | |
| Genetic reagent (*Mus musculus*) | Short: Rasgrf2-2A-dCre *B6;129S-Rasgrf2tm1(cre/folA)Hze/J* | *Harris et al., 2014* | RRID:JAX 022864 | |
| Genetic reagent (*Mus musculus*) | Short: Gpr26-Cre *Tg(Gpr26-cre) KO250Gsat/Mmucd* | *Gong et al., 2007* | RRID:MMRRC_033032-UCD | |
| Recombinant DNA reagent | AAV9.CaMKII.Flex.hChR2(H134R)-YFP.WPRE3 | Charité vector core | n/a | |

*Continued*

| Reagent type (species) or resource | Designation | Source or reference | Identifiers | Additional information |
|---|---|---|---|---|
| Recombinant DNA reagent | AAV1.Syn.Flex. GCaMP6s. WPRE.SV40 | Addgene | RRID:Addgene_100845 | |
| Chemical compound, drug | Cal-590 AM | AAT Bioquest | 20,510 | |
| Chemical compound, drug | Trimethoprim | Sigma-Aldrich | T7883 | |
| Software, algorithm | Matlab | Mathworks | RRID:SCR_001622 | |

## Animal models

Three transgenic mouse lines, Sim1-KJ18-Cre (*Tg(Sim1-cre)KJ18Gsat*) (*Gerfen et al., 2013*) × Ai32 (*Ai32(RCL-ChR2(H134R)/EYFP)*) (*Madisen et al., 2012*), Rasgrf2-2A-dCre (*B6;129S-Rasgrf2tm1(cre/folA)Hze/J*) and Gpr26-Cre (*Tg(Gpr26- cre)KO250Gsat/Mmucd*) were used in this study. The age of studied mice was P40–70. Both male and female mice were used. No food or water restriction was imposed. All studied mice were in good health. Mice were housed in single-sex groups in plastic cages with disposable bedding on a 12 hr light/dark cycle with food and water available ad libitum. Experiments were done during both dark and light phase. All procedures were approved and conducted in accordance with the guidelines given by the veterinary office of Landesamt für Gesundheit und Soziales Berlin (registration number G0278/16).

## Virus injection

Rasgraf2-2A-dCre and Gpr26-Cre mice were initially anesthetized with Isoflurane (1–2.5% in $O_2$ vol/vol, Abbott) before ketamine/xylazine anesthesia (75/10 mg/kg of body weight, respectively) was administered intraperitoneally. Lidocaine (1% wt/vol, Braun) was injected around the surgical site. Body temperature was maintained at ~36°C by a heating pad and the depth of anesthesia was monitored throughout the virus injection. Once anesthetized, the head was stabilized in a stereotaxic instrument (SR-5R, Narishige, Tokyo). The skull was exposed by a skin incision and a small hole (~0.5 × 0.5 mm²) was made above the primary motor cortex (1.0 mm anterior to bregma and 1.2 mm from midline) of Rasgraf2-2A-dCre mice or the POm (1.8 mm posterior to bregma and 1.25 mm from midline) of Gpr26-Cre mice. AAV9.CaMKII.Flex.hChR2(H134R)-YFP.WPRE3 (Charité Vector Core) or AAV1.Syn. Flex.GCaMP6s.WPRE.SV40 (Addgene #100845) was injected to Rasgrf2-2A-dCre or Gpr26-Cre mice, respectively. Each construct was backloaded into a glass micropipette (Drummond) and was slowly injected (at 20 ml/min, total amount 40–50 nl). The pipette remained there for another 2–5 min after injection. The skin was sutured after retracting the pipette. Rasgrf2-2A-dCre mice were subsequently injected with trimethoprim (TMP, 150 mg/g of body weight) intraperitoneally.

## Fabrication of double-µPeriscope

As described earlier (*Suzuki and Larkum, 2017*), single µPeriscopes were assembled in-house with a 0.10 × 0.10 mm² or 0.18 × 0.18 mm² micro right-angled prism (Edmund Optics), a 100 µm core multimode optical fiber (NA 0.22, Edmund Optics) and a custom-designed Grin lens (NA 0.2, Grin Tech) using a UV curable adhesive (Noland). Optical properties of single µPeriscope were characterized previously (*Suzuki and Larkum, 2017*).

The two µPeriscopes were precisely aligned with a specific distance between the two prisms and glued with a UV curable adhesive (Noland). Alternatively, two µPeriscopes were not glued but individually manipulated by two stereotaxic micromanipulators (SM-15R, Narishige).

## Extracellular recordings

Animals were initially anesthetized by isoflurane (1–2% in O2, vol/vol, Abbott) before urethane anesthesia (0.05 mg/kg of body weight) was administered intraperitoneally. Lidocaine (1%, wt/vol, Braun) was injected around the surgical site. Body temperature was maintained at ~36°C by a heating pad and the depth of anesthesia was monitored throughout experiment. Once anesthetized, the head was

stabilized in the stereotaxic instrument and the skull was exposed by a skin incision. A ~1.0 × 1.0 mm² craniotomy was made above the primary somatosensory (barrel) cortex and the dura matter was removed. The area was kept moist with rat ringer for the entire experiment (135 mM NaCl, 5.4 mM KCl, 1.8 mM $CaCl_2$, 1 mM $MgCl_2$, 5 mM HEPES(4-(2-hydroxyethyl)-1-piperazineethanesulfonic acid)). A linear array of 16 electrodes (NeuroNexus, A1 × 16-3 mm-50-177-A16) or a glass pipette was perpendicularly inserted into the area such that the uppermost electrode was positioned at 50 μm below the pia. Electrical activity was bandpass-filtered at 1–9 kHz, digitized at 10 kHz, amplified by ERP-27 system and Cheetah software (Neuralynx).

## Optogenetic stimulation with a double-μPeriscope

The end of each optical fiber was coupled with a blue LED (peak wavelength 470 nm, Cree or Thorlabs). The timing and intensity of optical stimulation through each μPeriscope were controlled by Power1401 and Spike two software (CED) and synchronized with the neural recording system or fluorescence imaging via TTL signals. The light intensity was 12 mW/mm² and the duration was 20 ms. Surgical preparation is same as described in the section 'Extracellular recordings'. A double-μPeriscope was slowly inserted into the somatosensory cortex such that the upper μPeriscope stimulated the distal apical dendrites (50–150 μm deep from pia) and the lower one stimulated the perisomatic region of L5 pyramidal neurons (500–700 μm deep from pia).

## Calcium imaging with a double-μPeriscope

The imaging setup consisted of a double-μPeriscope, an LED (peak wavelength 470 nm, Thorlabs), an excitation filter (480/30 bandpass, Chroma), an emission filter (535/40 bandpass, Chroma), a dichroic mirror (cutoff wavelength: 505 nm, Chroma), an 80 × 80 pixel high-speed CCD camera with frame rate of 125 Hz and 14-bit digitization (Redshirt Imaging), a 10× infinity corrected objective (58-372, Edmund Optics), and a tube lens (Optem, RL091301-1). Surgical preparation is same as described in the section 'Extracellular recordings'. An aluminum head implant was fixed to the skull of the mouse with dental cement and the mice were habituated to head-fixation and whisker deflection by a piezo element before imaging. A double- μPeriscope was slowly inserted into the barrel cortex such that the upper μPeriscope covered L1 (0–100 μm deep from pia) and the lower μPeriscope covered L5a (450–550 μm deep from pia). $\Delta F/F$ was calculated as $(F − F_0)/F_0$, where $F$ is the fluorescence intensity at any time point and $F_0$ is the average intensity over the prestimulus period of 320 ms. If $\Delta F/F$ is calculated as $(F − F_0)/(F_0 − F_b)$ where $F_b$ is the background fluorescence measured from a region away from the optical fiber, the scale bar in *Figure 2g* is equal to ~0.2% $\Delta F/F$.

## Calcium imaging combined with optogenetic stimulation

The imaging setup is same as described in the section 'Calcium imaging with a double-μPeriscope' except an LED (peak wavelength 565 nm, Thorlabs), an excitation filter (555/20 bandpass, AHF), an emission filter (605/55 bandpass, AHF) and a dichroic mirror (cutoff wavelength: 565 nm, AHF). Surgical preparation is same as described in the section 'Extracellular recordings'. An aluminum head implant was fixed to the skull of the mouse with dental cement and the mice were habituated to head-fixation before imaging. A double-μPeriscope was slowly inserted into the motor cortex so that the upper μPeriscope covered L2/3 (150–300 μm deep from pia) and the lower μPeriscope covered L5 (500–700 μm deep from pia). The calcium indicator Cal-590 AM (AAT Bioquest) was backloaded into a micropipette (Drummond) and slowly injected (at 20 nl/min, total 40–50 nl) to L5 (~600 μm below pia) of the primary motor cortex 1.5–2 hr before imaging experiments. The pipette remained there for at least 5 min after injection. $\Delta F/F$ was calculated in the same way as described in the section 'Calcium imaging with a double-μPeriscope'.

## Data analysis and statistical methods

Analyses were conducted using Matlab (Mathworks). Significance was determined by two-tailed, paired Student's *t*-test at a significance level of 0.05. The statistical test was chosen based on the data distribution using histogram. No statistical method was used to predetermine sample sizes, but our sample sizes are those generally employed in the field. We did not exclude any animal for data analysis. The variance was generally similar between groups under comparison. No blinding/randomization was performed.

## Acknowledgements

MS was supported by Deutsche Forschungsgemeinschaft (Exc 257 NeuroCure, project no. 327654276 SFB1315 TP.A04) and Research Foundation of Opto-Science and Technology. JA was supported by the European Regional Funds through the IT Academy Programme and the European Union's Horizon 2020 Research and Innovation Programme under the Marie Skłodowska-Curie grant agreement no. 799,411. MEL was supported by Deutsche Forschungsgemeinschaft (Exc 257 NeuroCure, project no. 327654276 SFB1315 TP.A04) and the European Union's Horizon 2020 research and innovation program under grant agreement no. 670,118 (ERC ActiveCortex).

## Additional information

### Funding

| Funder | Grant reference number | Author |
| --- | --- | --- |
| Deutsche Forschungsgemeinschaft | Exc 257 NeuroCure Project no. 327654276 SFB1315 TP.A04 | Mototaka Suzuki Matthew E Larkum |
| European Union's Horizon 2020 research and innovation program | Agreement no. 670118 (ERC ActiveCortex) | Matthew E Larkum |
| Marie Skłodowska-Curie Grant | Agreement no. 799411 | Jaan Aru |
| European Regional Funds through the IT Academy Programme | | Jaan Aru |
| Research Foundation for Opto-Science and Technology | | Mototaka Suzuki |

The funders had no role in study design, data collection, and interpretation, or the decision to submit the work for publication.

### Author contributions

Mototaka Suzuki, Conceptualization, Formal analysis, Funding acquisition, Visualization, Writing – original draft, Writing – review and editing; Jaan Aru, Formal analysis, Investigation, Writing – review and editing; Matthew E Larkum, Funding acquisition, Supervision, Writing – review and editing

### Author ORCIDs

Mototaka Suzuki (iD) http://orcid.org/0000-0002-2151-4882
Matthew E Larkum (iD) http://orcid.org/0000-0001-9799-2656

### Ethics

All procedures were approved and conducted in strict accordance with the guidelines given by the veterinary office of Landesamt für Gesundheit und Soziales Berlin (registration number G0278/16). Every effort was made to minimize suffering.

### Decision letter and Author response

Decision letter https://doi.org/10.7554/eLife.72894.sa1
Author response https://doi.org/10.7554/eLife.72894.sa2

## Additional files

### Supplementary files
• Transparent reporting form

## Data availability

Source data for Figures 1 & 2 are deposited in Dryad under DOI:https://doi.org/10.5061/dryad.crjdfn34z.

The following dataset was generated:

| Author(s) | Year | Dataset title | Dataset URL | Database and Identifier |
|---|---|---|---|---|
| Suzuki M, Aru J, Larkum ME | 2021 | Double-Periscope: a tool for multi-layer optical recordings, optogenetic stimulations or both | https://doi.org/10.5061/dryad.crjdfn34z | Dryad Digital Repository, 10.5061/dryad.crjdfn34z |

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
