## [Editor Report]

This Tools and Resources article details an innovative method for simultaneously stimulating and imaging two cortical layers in tandem while causing minimal damage to brain tissue. The method substantially builds on existing methods in several ways, while still pinpointing the limitations of existing methods that are overcome in this new approach. Three well-described sets of experiments demonstrate the method's reliability and versatility, and highlight its promise in tackling big questions about cortical microcircuit functions.

---

## [Decision Letter]

**Decision letter after peer review:**

Thank you for submitting your article "Double-µPeriscope: a tool for multi-layer optical recordings, optogenetic stimulations or both" for consideration by *eLife*. Your article has been reviewed by 2 peer reviewers, and the evaluation has been overseen by a Reviewing Editor and Tirin Moore as the Senior Editor. The reviewers have opted to remain anonymous.

The reviewers have discussed their reviews with one another, and the Reviewing Editor has drafted this to help you prepare a revised submission. As you will see from the detailed reviews, both reviewers agree that the described method is useful to the community. Nevertheless, they also note that the degree of innovation needs to be clarified before a final decision can be reached.

Essential revisions:

1. In fiber photometry, several papers and methods using multiple fibers to capture signals at multiple brain regions simultaneously have been published. For instance, see Kim C. et. al, "Simultaneous fast measurement of circuit dynamics at multiple sites across the mammalian brain" (2016) or Sych Y. et. al, "High-density multi-fiber photometry for studying large-scale brain circuit dynamics" (2019). In these papers, 5~48 regions are imaged simultaneously. These studies should be acknowledged and your method should be compared to these previous papers.

2. Adding a prism towards the end of the fiber is certainly an innovation, and enables layer specific imaging. However, as fiber photometry essentially measures the local population signal, even without the prism, layer specific signals could be measured, provided that the tip of the fiber lands on the appropriate layer and the excitation power is appropriate. Therefore, please provide additional information on how large or how local the brain region that each fiber tip can capture is. Furthermore, please relate this to the excitation power, fiber NA, etc. to better describe how your method improves or extends previous methods.

*Reviewer #1:*

The goal of this Tools and Resources article was to present a new method for optogenetic stimulation and optical imaging at the same time in two different cortical layers in vivo, and through 3 sets of experiments, highlight the promise and wide applicability of this method.

The method itself presents an elegant solution to several outstanding drawbacks among the many recent innovations in these lines of methodology, including high expense, lack of specificity and excessive brain tissue damage. The paper provides what I believe to be a fair account of the capabilities and limitations of existing methods and a clear description of how the new method builds on and overcomes these.

The three sets of experiments work well because they demonstrate reliability and feasibility in replicating previous findings from older techniques such as the phenomenon of 'backpropagation-activated calcium spike firing' and net inhibitory influence of layer 2/3 cells on layer 5 cells, while also extending beyond those findings by verifying that some effects generalise to other areas than previous observations – the layer 2/3-5 interaction previously seen in primary somatosensory is here extended to motor cortex – and uncovering interesting phenomena that are relatively unexplored to date – the great variability in the degree of mirroring of activity in two layers receiving axonal input from the same thalamic area.

The method presents exciting possibilities for the fine-grained study of cortical microcircuits and how they enable perception and cognition and relate to behaviour. The simplicity and low cost of the solution opens it up to a wider range of laboratories globally, and its low-profile imprint on the cortex ensures that it most likely reflects activity of normal, intact, rather than damaged, cortical tissue.

Comments for the authors:

A general comment is that the paper is very fast-paced and could, if possible, be fleshed out to a more deliberate pace, especially in pinpointing why the drawbacks of existing methods are really a problem, emphasising the importance and tantalising nature of the research questions addressed by the 3 sets of experiments, and in walking the reader through the figures.

To take examples:

– In the brief preamble to the 3rd experiment, it is stated that POm sends axons to layers 1 and 5a in S1, but it's unknown whether the activity of these pathways are the same or different. How MIGHT they be different and what would be the significance of that?

– At the end of the first paragraph in the introduction you could spell out more what is wrong with activity being complex – is it hard to disentangle distinct contributions from different neuronal structures? And what is bad about polysynaptic activations, and is this a problem caused by the extracellular signal recording (which by then has become the subject of the paragraph) or by the optogenetic stimulation? It will be obvious to some but not to a general audience. Similarly, what is bad about repairing and rewiring processes taking place? It's implicit for many readers but for the sake of a few more words you could spell out that this thwarts the measurement of representative, normal activity in the intact cortex.

Please state what a GRIN lens is.

The figure captions are extremely brief and could be expanded a little to guide the reader more in what to appreciate from each panel and to explain more of the labels and structures in the panels, even if it means slight repetition with the main text. For example, the acronyms in Figure 1 c could be explained, some further orientation could be provided for the photomicrograph in fig1d (e.g. indicate where the cortical surface is, and the layers), say what "Ex. Light" and "Em. Light" mean, etc. Even if most neuroscientists can figure it out, there is no harm in making it fully accessible to as wide an audience as possible.

On page 2 'this hypothesis' is referred to but none was stated.

What does it mean that the "spatial resolution of imaged structure is unavailable"?

It would be helpful, since there are so many experimental procedures, if the different sections in the methods were linked to specific results shown in the main paper, e.g. one heading could be "Extracellular recordings (Figure 1e)"

*Reviewer #2:*

This manuscript reported a new approach to conduct neural activity imaging and manipulation in two different cortical layers. Two periscopes, each constructed from a micro-prism, a GRIN lens and a multi-mode fiber, could be inserted to the brain at different depths, and each can either perform imaging or optogenetics. The authors demonstrated a few applications: stimulation of L5 soma and superficial layer dendrites to evoke backpropagating action potential; optogenetically stimulating cells in L2/3 and observing response in L5 to investigate interaction between cells in two different layers; and simultaneously recording axon terminals from posteromedial thalamic nucleus at two different depths in cortex. This works combines the ideas of fiber photometry to access deep layers and using microprism to turn the optical field of view by 90 deg.

Major strengths

• Using microprism to perform layer specific imaging or optogenetics.

• Low cost

• Demonstrations of a few applications that require layer specific imaging and optogenetics.

Major weakness

• As this is an inherently a variation of fiber photometry, there is a lack of cellular resolution and there is tissue damage.

• Innovation is modest, as it is an incremental improvement of fiber photometry. Some of the applications may be performed through regular fiber photometry as well.

• There is a lack of details on the optical setup and characterization of the periscope, i.e. how to choose the fiber, GRIN lens; optical throughput etc.

Overall, this research provides a new method to image/manipulate the neural activity of two different cortical layers. However, more details are needed on the optical setup and characterization of the periscope. The innovation of this work is modest.

Comments for the authors:

1. The df/f in Figure 2c and Figure 2g is very small. Does this represent a limitation or it is nature of the ensemble signal?

2. The assessment of two-photon microscope should be more fair. There have been various techniques in two-photon microscope that can image multiple depths together, and the depth range could be very large.

3. The lack of spatial resolution needs to be discussed and addressed as a limitation.

---

## [Author Response]

Essential revisions:1. In fiber photometry, several papers and methods using multiple fibers to capture signals at multiple brain regions simultaneously have been published. For instance, see Kim C. et. al, "Simultaneous fast measurement of circuit dynamics at multiple sites across the mammalian brain" (2016) or Sych Y. et. al, "High-density multi-fiber photometry for studying large-scale brain circuit dynamics" (2019). In these papers, 5~48 regions are imaged simultaneously. These studies should be acknowledged and your method should be compared to these previous papers.

We thank the reviewers and editors for this suggestion. Now we cite both papers in the Discussion and describe the advantages and target applications of our method. Bare optical fibers used in the above-mentioned papers are suitable if the target brain areas are sufficiently separate (> 1 mm), but our method has significant advantages to investigate more closely separate (< 1 mm) structures such as cortical layers in the rodent cortex. If one repeated the three experiments we described in the manuscript with two bare optical fibers, it would damage the cortical tissue more severely, would need to precisely calibrate the light power, and would be extremely difficult to get good signals.

2. Adding a prism towards the end of the fiber is certainly an innovation, and enables layer specific imaging. However, as fiber photometry essentially measures the local population signal, even without the prism, layer specific signals could be measured, provided that the tip of the fiber lands on the appropriate layer and the excitation power is appropriate. Therefore, please provide additional information on how large or how local the brain region that each fiber tip can capture is. Furthermore, please relate this to the excitation power, fiber NA, etc. to better describe how your method improves or extends previous methods.

Optical fibers with a flat end used in the above-mentioned papers would need higher pressure and therefore damage the cortical tissue upon insertion. In contrast, the sharp edge of microprism allows us to smoothly insert the probe with significantly less damage. On top of the damage, layer-specific activation/imaging with a downward oriented light beam is problematic; in principle it is possible to calibrate the light power to optimize the signal from a single layer versus the layers below, but in practice this is always an approximation and it is impossible to be sure in any given preparation exactly how much signal is derived from the desired layer versus the others. Furthermore, it involves reducing the power of the excitation light considerably which leads to signal-to-noise issues. We are unaware of any previous study that underwent these processes and claimed layer-specific activation/imaging with bare optical fibers only, implying that the approach is not fruitful. We have added a comment about these issues into the “Discussion”.

In addition to the micro-prism, the custom-designed gradient-index (GRIN) lens plays a key role in our method; it collimates light so that the optical stimulation is focal (i.e., light does not spread), as shown and calibrated in our earlier publication in 2017 (Figure 1b in http://doi.org/10.1038/s41467-01700282-4). If it were not for the GRIN lens, the light spreads (e.g., if fiber NA=0.22, acceptance angle=~25.4 deg) and is therefore less specific. As suggested by the reviewer, more detailed information (excitation power, NA etc) is now provided in “Methods”.

Reviewer #1:[…] A general comment is that the paper is very fast-paced and could, if possible, be fleshed out to a more deliberate pace, especially in pinpointing why the drawbacks of existing methods are really a problem, emphasising the importance and tantalising nature of the research questions addressed by the 3 sets of experiments, and in walking the reader through the figures.

Thank you for this suggestion. We have now added a considerable amount of new text to the introduction and discussion, highlighted some aspects of the research questions and better incorporated the explanation of figures into the text.

To take examples:– In the brief preamble to the 3rd experiment, it is stated that POm sends axons to layers 1 and 5a in S1, but it's unknown whether the activity of these pathways are the same or different. How MIGHT they be different and what would be the significance of that?

An implicitly-held yet prevailing assumption is that two axonal terminals are collateral (i.e., branching axons from same cells). However, due to technical difficulties, this assumption has never been directly tested. Surprisingly, our data obtained with the new tool appear to challenge the assumption. This result would imply that higher-order thalamus sends different information to layer 1 and layer 5a, a result that would actually be consistent with our recent finding that the coupling across the apical axis of deep pyramidal neurons is controlled by higher-order thalamus due to metabotropic receptor activation approximately in layer 5a (Suzuki and Larkum, 2020; Aru et al., 2020). The result, if thoroughly confirmed, would be important and possibly astonishing for many neuroscientists. Although confirming the finding requires a lot more experiments, which are out of the scope of this “Tools and Resources” paper, the point we would like to make here is that such an inexpensive tool allows us to directly examine long-unexplored assumptions in the brain of awake behaving animals. We elaborate on this point in the manuscript.

– At the end of the first paragraph in the introduction you could spell out more what is wrong with activity being complex – is it hard to disentangle distinct contributions from different neuronal structures? And what is bad about polysynaptic activations, and is this a problem caused by the extracellular signal recording (which by then has become the subject of the paragraph) or by the optogenetic stimulation? It will be obvious to some but not to a general audience. Similarly, what is bad about repairing and rewiring processes taking place? It's implicit for many readers but for the sake of a few more words you could spell out that this thwarts the measurement of representative, normal activity in the intact cortex.

We thank Reviewer #1 for these suggestions. We now better explain these in “Introduction”.

Please state what a GRIN lens is.

We now explain the acronym (gradient-index lens).

The figure captions are extremely brief and could be expanded a little to guide the reader more in what to appreciate from each panel and to explain more of the labels and structures in the panels, even if it means slight repetition with the main text. For example, the acronyms in Figure 1 c could be explained, some further orientation could be provided for the photomicrograph in fig1d (e.g. indicate where the cortical surface is, and the layers), say what "Ex. Light" and "Em. Light" mean, etc. Even if most neuroscientists can figure it out, there is no harm in making it fully accessible to as wide an audience as possible.

We have now expanded the figure captions and explained the acronyms*.*

On page 2 'this hypothesis' is referred to but none was stated.

We rephrased the sentence.

What does it mean that the "spatial resolution of imaged structure is unavailable"?

We meant that since in the present design each µPeriscope uses one multi-mode optical fiber, the signal we get is the summation of fluorescence from all stimulated structures (e.g., cell bodies, dendrites) in a specific layer. In applications that need to visualize subcellular structures, a high density bundle fiber can be used. We rephrased the corresponding sentence.

It would be helpful, since there are so many experimental procedures, if the different sections in the methods were linked to specific results shown in the main paper, e.g. one heading could be "Extracellular recordings (Figure 1e)"

We linked methods subsections to the corresponding figure panels as the reviewer suggested.

Reviewer #2:[…] 1. The df/f in Figure 2c and Figure 2g is very small. Does this represent a limitation or it is nature of the ensemble signal?

This does not represent a limitation. Figure 2g shows clear traces of axonal calcium signal in each layer on a single trial. The small ΔF/F value is due to a deliberately simplified method to calculate ΔF/F; as described in “Materials and methods” section, “ΔF/F was calculated as (F–F0)/F0, where F is the fluorescence intensity at any time point and F0 is the average intensity over the pre-stimulus period of 320 ms.” Since each pixel value is 14-bit, ΔF/F becomes very small with this method. If ΔF/F is calculated as (F-F0)/(F0-Fb) where Fb is the background fluorescence measured from a region away from the optical fiber, the scale bar in Figure 2g becomes ~0.2% ΔF/F. Therefore, ΔF/F greatly changes depending on the calculation method. To help readers understand the small ΔF/F values, we explained the reason in “Materials and methods”. We thank Reviewer #2 for pointing this out.

2. The assessment of two-photon microscope should be more fair. There have been various techniques in two-photon microscope that can image multiple depths together, and the depth range could be very large.

Now we additionally cite papers reporting the development of fast scanning methods by spatiotemporal multiplexing of excitation light. However, due to technical difficulties and expensiveness, few laboratories can easily replicate such systems. Additionally, we would like to note that two-photon imaging of axonal activities is still challenging, even at a single focal plane (see https://doi.org/10.1038/s41593-018-0211-4), especially if the depth is > 500 µm. Therefore, fast axonal imaging at multiple depths is even more challenging. Our approach that inserts a micro-optical probe can reliably measure the axonal activity (Figure 2g) with a commercially available CCD or CMOS camera which is an order of magnitude less expensive.

3. The lack of spatial resolution needs to be discussed and addressed as a limitation.

The three experiments we presented did not need spatial resolution, but when necessary, using a high density optical fiber bundle would provide spatial subcellular resolution. We rephrased the corresponding sentence in “Discussion”.

We thank the 2 reviewers and editors for all these constructive comments that have significantly improved our manuscript.